# Maternal smoking DNA methylation risk score associated with health outcomes in offspring of European and South Asian ancestry

Wei Q Deng[1,2,3]*, Nathan Cawte[4], Natalie Campbell[1], Sandi M Azab[1,5], Russell J de Souza[1,5], Amel Lamri[1,4], Katherine M Morrison[6], Stephanie A Atkinson[6], Padmaja Subbarao[7], Stuart E Turvey[8], Theo J Moraes[7,9], Koon K Teo[1,4,5], Piush J Mandhane[10], Meghan B Azad[11], Elinor Simons[12], Guillaume Paré[4,5,13,14], Sonia S Anand[1,4,5]*

[1]Department of Medicine, Faculty of Health Sciences, McMaster University, Hamilton, Canada; [2]Peter Boris Centre for Addictions Research, St. Joseph's Healthcare Hamilton, Hamilton, Canada; [3]Department of Psychiatry and Behavioural Neurosciences, McMaster University, Hamilton, Canada; [4]Population Health Research Institute, David Braley Cardiac, Vascular and Stroke Research Institute, Hamilton, Canada; [5]Department of Health Research Methods, Evidence, and Impact, McMaster University, Hamilton, Canada; [6]Department of Pediatrics, McMaster University, Hamilton, Canada; [7]Department of Pediatrics, University of Toronto, Toronto, Canada; [8]Department of Pediatrics, BC Children's Hospital, The University of British Columbia, Vancouver, Canada; [9]Program in Translational Medicine, SickKids Research Institute, Toronto, Canada; [10]Department of Pediatrics, University of Alberta, Edmonton, Canada; [11]Children's Hospital Research Institute of Manitoba, Department of Pediatrics and Child Health, University of Manitoba, Winnipeg, Canada; [12]Section of Allergy and Immunology, Department of Pediatrics and Child Health, University of Manitoba, Manitoba, Canada; [13]Thrombosis and Atherosclerosis Research Institute, David Braley Cardiac, Vascular and Stroke Research Institute, Hamilton, Canada; [14]Department of Pathology and Molecular Medicine, McMaster University, Michael G. DeGroote School of Medicine, Hamilton, Canada

*For correspondence:
dengwq@mcmaster.ca (WQD);
anands@mcmaster.ca (SSA)

**Competing interest:** The authors declare that no competing interests exist.

## Abstract

**Background:** Maternal smoking has been linked to adverse health outcomes in newborns but the extent to which it impacts newborn health has not been quantified through an aggregated cord blood DNA methylation (DNAm) score. Here, we examine the feasibility of using cord blood DNAm scores leveraging large external studies as discovery samples to capture the epigenetic signature of maternal smoking and its influence on newborns in White European and South Asian populations.

**Methods:** We first examined the association between individual CpGs and cigarette smoking during pregnancy, and smoking exposure in two White European birth cohorts (n=744). Leveraging established CpGs for maternal smoking, we constructed a cord blood epigenetic score of maternal smoking that was validated in one of the European-origin cohorts (n=347). This score was then tested for association with smoking status, secondary smoking exposure during pregnancy, and health outcomes in offspring measured after birth in an independent White European (n=397) and a South Asian birth cohort (n=504).

**Results:** Several previously reported genes for maternal smoking were supported, with the strongest and most consistent association signal from the *GFI1* gene (6 CpGs with p<5 × 10⁻⁵). The epigenetic maternal smoking score was strongly associated with smoking status during pregnancy (OR = 1.09 [1.07, 1.10], p=5.5 × 10⁻³³) and more hours of self-reported smoking exposure per week (1.93 [1.27, 2.58], p=7.8 × 10⁻⁹) in White Europeans. However, it was not associated with self-reported exposure (p>0.05) among South Asians, likely due to a lack of smoking in this group. The same score was consistently associated with a smaller birth size (–0.37±0.12 cm, p=0.0023) in the South Asian cohort and a lower birth weight (–0.043±0.013 kg, p=0.0011) in the combined cohorts.

**Conclusions:** This cord blood epigenetic score can help identify babies exposed to maternal smoking and assess its long-term impact on growth. Notably, these results indicate a consistent association between the DNAm signature of maternal smoking and a small body size and low birth weight in newborns, in both White European mothers who exhibited some amount of smoking and in South Asian mothers who themselves were not active smokers.

**Funding:** This study was funded by the Canadian Institutes of Health Research Metabolomics Team Grant: MWG-146332.

## eLife assessment

This study offers a **useful** advance by introducing a cord blood DNA methylation score for maternal smoking effects, with the inclusion of cohorts from diverse backgrounds. However, the overall strength of evidence is deemed **incomplete**, due to concerns regarding low exposure levels and low statistical power, which hampers the generalisability of their findings. The study provides an interesting basis for future studies, but would benefit from the addition of more cohorts to validate the findings and a focus on more diverse health outcomes.

## Introduction

Maternal smoking has adverse effects on offspring health including pre-term delivery (*Stock and Bauld, 2020*; *Liu et al., 2020*), stillbirth (*Marufu et al., 2015*), and low birth weight (*Ventura et al., 2003*), and is associated with pregnancy complications such as maternal higher blood pressure, and gestational diabetes (*National Center for Chronic Disease Prevention and Health Promotion (US) Office on Smoking and Health, 2014*). Consistent with the Developmental Origins of Health and Disease (DOHaD) hypothesis, maternal smoking exposes the developing fetus to harmful chemicals in tobacco that negatively impact the health of newborns, resulting in early-onset metabolic diseases, such as childhood obesity (*Montgomery and Ekbom, 2002*; *Toschke et al., 2002*; *Oken et al., 2008*; *Philips et al., 2020*). Yet self-reported smoking status is subject to underreporting among pregnant women (*England et al., 2007*; *Shipton et al., 2009*; *Salmasi et al., 2010*). This could subsequently impact the effectiveness of interventions aimed at reducing smoking during pregnancy and may skew data on the risks associated with maternal smoking.

DNA methylation is one of the most commonly studied epigenetic mechanisms by which cells regulate gene expression, and is increasingly recognized for its potential as a biomarker (*Yousefi et al., 2022*). Differential DNA methylation has been established as a reliable biochemical response to cigarette smoking and was shown to capture the long-lasting effects of persistent smoking in ex-smokers (*Shenker et al., 2013*; *Joehanes et al., 2016*; *Guida et al., 2015*). Recent large epigenome-wide association studies (EWAS) have robustly identified differentially methylated cytosine–phosphate–guanine (CpG) sites associated with adult smoking (*Joehanes et al., 2016*; *Sikdar et al., 2019*; *Zeilinger et al., 2013*) and maternal smoking (*Joubert et al., 2016*; *Hannon et al., 2019*). Our recent systematic review of 17 cord blood EWAS found that out of the 290 CpG sites reported to be associated with at least one of the following: maternal diabetes, pre-pregnancy body mass index (BMI), diet during pregnancy, smoking, and gestational age, 19 sites were identified in more than one study and all of them associated with maternal smoking (*Akhabir et al., 2022*). Furthermore, these findings have led to a more thorough investigation of the epigenetic mechanisms underlying associations between well-established epidemiological exposures and outcomes, such as the relationship between maternal smoking and birth weight in Europeans (*Hannon et al., 2019*; *Witt et al., 2018*; *Küpers et al., 2015*;

*Xu et al., 2021*; *Cardenas et al., 2019*) and the less studied African American populations (*Xu et al., 2021*) as well as between maternal diet and cardiovascular health (*Murray et al., 2021*).

Only a handful of cohort studies were designed to assess the influence of maternal exposures on DNA methylation changes in non-Europeans (*Xu et al., 2021*; *Raynor and Born in Bradford Collaborative Group, 2008*). It has been suggested that systematic patterns of methylation (*Elliott et al., 2022*), such as cell composition, could differ between individuals of different ancestral backgrounds, which could in turn confound the association between differential DNAm and smoking behaviors (*Choquet et al., 2021*). These systematic differences also contribute to different smoking-related methylation signals at individual CpGs (*Elliott et al., 2014*). Thus, a comparative study of maternal smoking exposure is a first step towards generalizing existing EWAS results to other populations and a necessary step towards addressing health disparities that exist between populations due to societal privilege, including race or ethnicity and socioeconomic factors.

A promising direction in epigenetic studies of adult smoking is the application of a methylation score *Bollepalli et al., 2019*; this strategy can also be applied to disseminate current knowledge on differential DNA methylation studies of maternal smoking. A methylation score is usually tissue-specific and combines information from multiple CpGs using statistical models (*Yousefi et al., 2022*). Reducing the number of predictors and measurement noise in the data can lead to better statistical power and a more parsimonious instrument for subsequent analyses. It is also of interest to determine whether methylation scores demonstrate the capacity to predict outcomes in diverse human populations, given the presence of systematic differences in methylation patterns due to ancestral backgrounds (*Elliott et al., 2022*).

In this paper, we investigated the epigenetic signature of maternal smoking on cord blood DNA methylation in newborns, as well as its association with newborn and later life outcomes in one South Asian which refers to people who originate from the Indian subcontinent, and two predominantly European-origin birth cohorts. Similar to the Born in Bradford study (*Wright et al., 2013*), we observed several differentiating epidemiological characteristics between South Asian and European-origin mothers. Notably, almost none of the South Asian mothers were current smokers and had low smoking rates pre-pregnancy as compared to European mothers, which is consistent with the broader trends of lower smoking rates in South Asian females (*Reitsma et al., 2021*). Another relevant observation is the small birth size and low birth weight in the South Asian newborns. These differences in newborn size and weight may be influenced by various factors, including maternal nutrition, genetics, and socioeconomic status. Keeping these differences in mind, we first conducted cohort-specific epigenetic association studies between available CpGs and maternal smoking in the predominantly European-origin cohorts, benchmarking with previously identified CpGs for maternal smoking and adult smoking. Second, we leveraged the reported summary statistics from existing large EWASs to construct a methylation risk score (MRS) for maternal smoking. The MRS was first internally validated in one of the European-origin cohorts and then tested in a second independent European-origin cohort. Third, we examined the association between maternal smoking MRS and newborn health outcomes, including length, weight, BMI ponderal index, and early-life anthropometrics in both European and South Asian cohorts.

## Materials and methods
### Study population

The NutriGen Alliance is a consortium consisting of four prospective, population-based birth cohorts that enrolled birthing mother and newborn pairs in Canada. Details of these cohorts have been described elsewhere (*de Souza et al., 2016*). The current investigation focused on (i). European-origin offspring from the population-based CHILD study who were selected for methylation analysis, (ii). The Family Atherosclerosis Monitoring In early life (FAMILY) study that is predominately European-origin, and (iii). The SouTh Asian biRth cohorT (START) study that is exclusively comprised of people who originated from the Indian subcontinent known as South Asians. The ethnicity of the parents was self-reported and recorded at baseline in all three cohorts. Biological samples, clinical assessments, and questionnaires were used to derive health phenotypes and an array of genetic, epigenetic, and metabolomic data. The superordinate goal of the NutriGen study is to understand how nutrition,

environmental exposures, and physical health of mothers impact the health and early development of their offspring using a multi-omics approach.

## Methylation data processing and quality controls

Newborn cord blood samples were processed using two methylation array technologies. About half of the START samples and selected samples from CHILD were hybridized to the Illumina Human-Methylation450K BeadChip (HM450K) array, which covers CpGs in the entire genome (*Bibikova et al., 2011*) The raw methylation data were generated by the Illumina iScan software and separately pre-processed for START and CHILD using the '*sesame*' R package following pipelines designed for HM450K BeadChip (*Zhou et al., 2018*). The FAMILY samples were profiled using a targeted array based on the Infinium Methylation EPIC designed by the Genetic and Molecular Epidemiology Laboratory (GMEL; Hamilton, Canada). The GMEL customized array includes ~3000 CpG sites that were previously reported to associate with complex traits or exposures and was designed to maximize discovery while keeping the costs of profiling epigenome-wide DNA methylation down. The targeted methylation data were pre-processed using a customized quality control pipeline and functions from the '*sesame*' R package (*Wanding Zhou, 2018*) recommended for EPIC.

Pre-processed data were then used to derive the β-value matrix, where each column gives the methylation level at a CpG site as a ratio of the probe intensity to the overall probe intensity. Additional quality control filters were applied to the final beta-value matrices to remove samples with >10% missing probes and CpG probes with >10% samples missing. Cross-reactive probes and SNP probes were removed as recommended for HM450 (*Chen et al., 2013*) and EPIC arrays (*Zhou et al., 2017*; *Pidsley et al., 2016*). For CpG probes with a missing rate <10%, mean imputation was used to fill in the missing values. We further excluded samples that were either mismatches between reported sex and methylation-inferred sex or were duplicates. Finally, considering the low prevalence of smokers, we sought to reduce spurious associations by removing non-informative probes that were either all hypomethylated (β-value<0.1) or hypermethylated (β-value>0.9), which have been shown to have less optimal performance (*Hillary et al., 2022*). A summary of the sample and probe inclusion/exclusion is shown in *Supplementary file 1a*.

Cell-type proportions (CD8T, CD4T, Natural Killer cells, B cells, monocytes, granulocytes, and nucleated red blood cells) were estimated following a reference-based approach developed for cord blood (*Gervin et al., 2019*) and using R packages '*FlowSorted.CordBloodCombined.450k*' and '*FlowSorted.Blood.EPIC.*' All data processing and subsequent analyses were conducted in R v.4.1.0 (*R Development Core Team, 2021*).

## Phenotype data processing and quality controls

At the time of enrollment, all pregnant women completed a comprehensive questionnaire that collected information on prenatal diet, smoking, education, socioeconomic factors, physical activities, and health as detailed previously (*Morrison et al., 2009*; *Anand et al., 2013*). Maternal smoking history (0=never smoked, 1=quit before this pregnancy, 2=quit during this pregnancy, or 3=current smoker) was assessed during the second trimester (at baseline). Smoke exposure was measured as the 'number of hours exposed per week.' GDM was determined based on a combination of oral glucose tolerance test (OGTT), self-report, and reported diabetic treatments (insulin, pills, or restricted diet). For South Asian mothers in START, the same OGTT threshold as Born in Bradford (*Raynor and Born in Bradford Collaborative Group, 2008*; *Wright et al., 2013*) was used, while the International Association of the Diabetes and Pregnancy Study Groups (IASDPSG) criteria (*Metzger et al., 2010*) for OGTT were used in CHILD and FAMILY cohorts. Mode of delivery (emergency c-section vs. other) was collected at the time of delivery.

Newborn length and weight were collected immediately after birth and extracted from the medical chart. The newborns were then followed up at 1, 2, 3, and 5 years of age and provided basic anthropometric measurements, including height, weight, hip and waist circumference, BMI, and sum of the skinfolds (triceps skinfold and subscapular skinfold). Additional phenotypes included smoking exposures (hours per week) at home, potential allergy based on the mother reporting any of: eczema, hay fever, wheezing , asthma, food allergy (egg, cow milk, soy, other) for her child in FAMILY and START, and asthma based on mother's opinion in CHILD ('In your opinion, does the child have any of the following? Asthma').

## Phenotype and methylation data consolidation

The current investigation examines the impact of maternal smoking or smoke exposure on DNA methylation derived from newborn cord blood in START and the two predominately European cohorts (CHILD and FAMILY). To maximize sample size in FAMILY and CHILD, we retained either self-identified or genetically confirmed Europeans based on available genetic data (*Supplementary file 1a*). The cohorts consist of representative population samples without enrichment for any clinical conditions, though only singleton mothers were invited to participate. For continuous phenotypes, an analysis of variance (ANOVA) using the *F*-statistics or a two-sample *t*-test was used to compare the mean difference across the three cohorts or two groups, respectively. For categorical phenotypes, a chi-square test of independence was used to compare the differences in frequencies of observed categories. Note that three of the categories under smoking history in the START cohort had expected cell counts less than 5, and was thus excluded from the comparison, the reported *p*-value was for CHILD and FAMILY.

The final analytical datasets, after combining the quality-controlled methylation data and phenotypic data, included 352, 411, and 504 mother-newborn pairs from CHILD, FAMILY, and START, respectively. Demographic characteristics and relevant covariates of the epigenetic subsample and the overall sample are summarized in *Table 1* and *Supplementary file 1b*, respectively.

## Epigenome-wide association of maternal smoking in European cohorts

Since there were no current smokers in START (*Table 1*), we tested the association between maternal smoking and differential methylated sites in FAMILY (# of CpG = 2544) and CHILD (# of CpG = 200,050). The primary outcome variable was 'current smoker,' defined by mothers self-identified as currently smoking during the pregnancy vs. those who never smoked or quit either before or during pregnancy. We also included a secondary outcome variable 'ever smoker,' defined by mothers who are current smokers or have quit smoking vs. those who never smoked. A tertiary outcome was smoking exposure, measured by the number of hours per week reported by the expectant mothers, and was available in all cohorts. We summarized the type of analyses for different outcomes in *Supplementary file 1c*.

We first conducted a separate epigenetic association study in each cohort, testing the association between methylation β-values at individual CpGs and the smoking phenotype using either a logistic regression model for smoking status or a linear regression for smoking exposure as the outcome. The model adjusted for additional covariates including the estimated cord blood cell proportions, maternal age, social disadvantage index, which is a continuous composite measure of social and economic exposures (*Anand et al., 2006*), mother's years of education, GDM, and parity. The smoking exposure variable was skewed, and a rank-based transformation was applied to mimic a standard normal distribution.

We then meta-analyzed association results for maternal smoking status in the European cohorts using an inverse variance-weighted fixed-effect model. The meta-analysis was conducted for 2,112 CpGs that were available in both CHILD (profiled using HM450K) and FAMILY (profiled using the targeted array). For the tertiary outcome, we conducted an inverse variance meta-analysis including START using both a fixed-effect model. For each EWAS or meta-analysis, the false discovery rate (FDR) adjustment was used to control multiple testing and we considered CpGs that passed an FDR-adjusted p-value <0.05 to be relevant for maternal smoking.

## Using DNA methylation to construct predictive models for maternal smoking

We sought to construct a predictive model in the form of a methylation risk score (MRS) using reported associations of maternal smoking. The proposed solution adapted the existing lassosum method (*Mak et al., 2017*) that was originally designed for polygenic risk scores, where the matrix of SNP genotypes (*X*) can be conveniently replaced by the β-value matrix. We hope to establish a linear regression model that can explain the variation in y using linear combinations of X:

$$y = X\gamma + \varepsilon, \tag{1}$$

**Table 1.** Characteristics of the epigenetic subsample (1267 mother–newborn pairs) from the CHILD, FAMILY, START cohorts.

| | Phenotypes | CHILD (n=352) | FAMILY (n=411) | START (n=504) | ANOVA F-test or Chi-squared test p-value for differences |
|---|---|---|---|---|---|
| Mother | Smoking History | | | | |
| | never smoked | 247 (70.2%) | 253 (61.6%) | 501 (99.4%) | <0.001* |
| | quit before this pregnancy | 72 (20.5%) | 58 (14.1%) | 1 (0.2%) | |
| | quit during this pregnancy | 17 (4.8%) | 57 (13.9%) | 1 (0.2%) | |
| | currently smoking | 11 (3.1%) | 29 (7.1%) | 0 (0%) | |
| | Missing | 5 (1.4%) | 14 (3.4%) | 1 (0.2%) | |
| | **Smoking Exposure (hr/week)** | | | | |
| | Mean (SD) | 0.97 (±7.64) | 2.52 (±12.83) | 0.33 (±2.67) | <0.001 |
| | Missing | 12 (3.4%) | 5 (1.2%) | 42 (8.3%) | |
| | **Gestational Diabetes Mellitus** | | | | |
| | YES | 16 (4.5%) | 66 (16.1%) | 183 (36.3%) | <0.001 |
| | NO | 336 (95.5%) | 345 (83.9%) | 320 (63.5%) | |
| | Missing | 0 (0%) | 0 (0%) | 1 (0.2%) | |
| | **Years of Education** | | | | <0.001 |
| | Mean (SD) | 16.96 (±3.08) | 16.85 (±3.39) | 15.81 (±2.41) | |
| | Missing | 7 (2.0%) | 3 (0.7%) | 0 (0%) | |
| | **Mother's Age** | | | | |
| | Mean (SD) | 32.69 (±4.45) | 31.86 (±5.42) | 30.12 (±3.91) | <0.001 |
| | Missing | 4 (1.1%) | 0 (0%) | 0 (0%) | |
| | **Parity** | | | | |
| | Mean (SD) | 0.72 (±0.88) | 0.80 (±1.02) | 0.80 (±0.81) | 0.098 |
| | Missing | 2 (0.6%) | 0 (0%) | 13 (2.6%) | |
| | **Pre-pregnancy BMI (kg/m2)** | | | | |
| | Mean (SD) | 24.78 (±5.42) | 26.46 (±6.38) | 23.71 (±4.45) | <0.001 |
| | Missing | 132 (37.5%) | 16 (3.9%) | 2 (0.4%) | |
| | **Newborn Sex** | | | | |
| | Male | 194 (55.1%) | 211 (51.3%) | 239 (47.4%) | 0.083 |
| | Female | 158 (44.9%) | 200 (48.7%) | 265 (52.6%) | |
| | **Plant-Based Diet** | | | | |
| | Mean (SD) | −0.48 (±0.46) | 0.19 (±0.67) | 1.56 (±1.14) | <0.001 |
| | Missing | 23 (6.5%) | 36 (8.8%) | 16 (3.2%) | |
| | **Health Conscious Diet** | | | | |
| | Mean (SD) | 0.21 (±0.81) | −0.73 (±0.73) | −0.42 (±0.79) | <0.001 |
| | Missing | 23 (6.5%) | 36 (8.8%) | 16 (3.2%) | |
| | **Western Diet** | | | | |
| | Mean (SD) | −0.15 (±0.63) | 1.06 (±1.20) | −0.51 (±0.65) | <0.001 |
| | Missing | 23 (6.5%) | 36 (8.8%) | 16 (3.2%) | |

*Table 1 continued on next page*

*Table 1 continued*

|  | Phenotypes | CHILD (n=352) | FAMILY (n=411) | START (n=504) | ANOVA F-test or Chi-squared test p-value for differences |
|---|---|---|---|---|---|
| Newborn | Gestational Age (weeks) |  |  |  |  |
|  | Mean (SD) | 39.53 (±1.38) | 39.44 (±1.47) | 39.20 (±1.32) | <0.001 |
|  | Missing | 4 (1.1%) | 0 (0%) | 0 (0%) |  |
|  | Birth Length (cm) |  |  |  |  |
|  | Mean (SD) | 51.68 (±2.52) | 50.20 (±2.16) | 51.44 (±2.69) | <0.001 |
|  | Missing | 71 (20.2%) | 10 (2.4%) | 7 (1.4%) |  |
|  | Birth Weight (kg) |  |  |  |  |
|  | Mean (SD) | 3.50 (±0.49) | 3.53 (±0.50) | 3.26 (±0.46) | <0.001 |
|  | Missing | 6 (1.7%) | 0 (0%) | 1 (0.2%) |  |
|  | Newborn BMI (kg/m2) |  |  |  |  |
|  | Mean (SD) | 13.11 (±1.41) | 13.94 (±1.29) | 12.31 (±1.39) | <0.001 |
|  | Missing | 72 (20.5%) | 10 (2.4%) | 7 (1.4%) |  |
|  | Newborn Ponderal Index (kg/m3) |  |  |  |  |
|  | Mean (SD) | 25.45 (±3.14) | 27.79 (±2.55) | 24.02 (±3.17) | <0.001 |
|  | Missing | 72 (20.5%) | 10 (2.4%) | 7 (1.4%) |  |
| Estimated cell proportions | CD8T |  |  |  |  |
|  | Mean (SD) | 0.01 (±0.01) | 0.04 (±0.03) | 0.02 (±0.02) | <0.001 |
|  | CD4T |  |  |  |  |
|  | Mean (SD) | 0.11 (±0.06) | 0.13 (±0.06) | 0.16 (±0.07) | <0.001 |
|  | NK |  |  |  |  |
|  | Mean (SD) | 0.02 (±0.02) | 0.03 (±0.03) | 0.02 (±0.03) | <0.001 |
|  | Bcell |  |  |  |  |
|  | Mean (SD) | 0.02 (±0.02) | 0.04 (±0.03) | 0.04 (±0.03) | <0.001 |
|  | Mono |  |  |  |  |
|  | Mean (SD) | 0.01 (±0.02) | 0.04 (±0.03) | 0.03 (±0.03) | <0.001 |
|  | Gran |  |  |  |  |
|  | Mean (SD) | 0.80 (±0.10) | 0.60 (±0.13) | 0.72 (±0.14) | <0.001 |
|  | nRBC |  |  |  |  |
|  | Mean (SD) | 0.08 (±0.08) | 0.12 (±0.11) | 0.07 (±0.11) | <0.001 |
|  | MNLR |  |  |  |  |
|  | Mean (SD) | 6.59 (±6.00) | 3.30 (±3.14) | 3.98 (±3.08) | <0.001 |
|  | Missing | 6 (1.7%) | 0 (0%) | 3 (0.6%) |  |
|  | * comparison for CHILD and FAMILY only |  |  |  |  |

where $X \in R^{n \times p}$ denotes a column standardized β-matrix of the $p$ CpGs measured on $n$ individuals. Multi-collinearity arises as many of the CpGs in physical proximity are highly correlated, causing instability in the model converging to a solution and/or leading to variance inflation in the resulting coefficients when estimated simultaneously. A lasso solution was designed to alleviate the multi-collinearity

of this estimation problem and can be obtained by minimizing the objective function that includes an L-1 penalty term that regularizes γ, forcing some of the coefficients to be exactly zero:

$$\hat{\gamma} = min_\gamma \left\{ n(y - X\gamma)^T(y - X\gamma) + 2\lambda \Sigma_{j=1}^{p} \left| \gamma_j \right| \right\} \tag{2}$$

Briefly, an objective function under elastic-net constraint was minimized to obtain the elastic-net solution γ, where only summary statistics (*b*) and a scalar of the covariance between the β-values of the CpGs ($X'X$) are needed. This was done by modifying the elastic net solution (https://github.com/tshmak/lassosum/blob/master/R/elnetR.R; *Mak, 2017*) for the lassosum method (*Mak et al., 2017*) that depended on two tuning parameters, along with additional inputs, namely the summary statistics and a reference CpG data covariance matrix. The Elastic net using the summary statistics function contained hyperparameters for the L-1 and L-2 penalty, namely, $\lambda_1$ and $\lambda_2$, which needed to be selected. To select the optimal tuning parameters, we examined a range of $\lambda_1$ values that forces all weights to be zero or no penalty, with 50 incremental increases, and $\lambda_2$ was taken to be α(1 − $\lambda_1$) where α was set to be 0–1 with incremental increases of 0.1. These together gave a grid of 10×50 choices for the two tuning parameter values. The tuning parameter pair that produced a score that was most significantly associated with the smoking history variable history (as a continuous outcome) in CHILD, without any data transformation, was chosen as the final elastic net solution. The optimized $\lambda_1$ and $\lambda_2$ were then used to create a final model that entails a list of CpGs and their corresponding weights, which were then used to calculate an MRS for maternal smoking in the FAMILY and START samples.

The summary statistics of the discovery of EWAS were obtained from the EWAS catalog (http://www.ewascatalog.org/) reported under 'PubMed ID 27040690' by Joubert and colleagues (*Joubert et al., 2016*). The summary statistics were restricted to the analysis of 'sustained maternal smoking in pregnancy effect on newborns adjusted for cell composition.' Of the 2620 maternal smoking CpGs that passed the initial screening, 2107 were available in CHILD but only 128 were common to CHILD, FAMILY, and START. To evaluate whether the targeted GMEL-EPIC array design has comparable performance as the epigenome-wide array to evaluate the epigenetic signature of maternal smoking, a total of three MRSs were constructed, two using the 128 CpGs available in all cohorts – across the HM450K and targeted GMEL-EPIC arrays – and with either CHILD (n=347 with non-missing smoking history) or FAMILY (n=397 with non-missing smoking history) as the validation cohort, and another using 2107 CpGs that were only available in CHILD and START samples with CHILD as the validation cohort. The validation model considered the continuous smoking history without modification as the outcome, while accounting for covariates, which included the estimated cord blood cell proportions, maternal age, social disadvantage index, mother's years of education, GDM, and parity. Henceforth, we referred to these derived maternal smoking scores as the FAMILY-targeted MRS, CHILD-targeted MRS, and the HM450K MRS, respectively. To benchmark and compare with existing maternal smoking MRSs, we calculated the Reese score using 28 CpGs (*Reese et al., 2017*; *Richmond et al., 2018*), Richmond score using 568 CpGs (*Richmond et al., 2018*), Rauschert score using 204 CpGs (*Rauschert et al., 2019*), Joubert score using all 2,620 CpGs with evidence of association for maternal smoking (*Joubert et al., 2016*), and finally a three-CpG score for air pollution (*Gondalia et al., 2019*). The details of these scores and score weight can be found in *Supplementary file 1d*.

## Statistical analysis

For each cohort, we contrasted the three versions of the derived scores using an analysis of variance analysis (ANOVA) along with pairwise comparisons using a two-sample *t*-test to examine how much information might be lost due to the exclusion of more than 10-fold CpGs at the validation stage, in all samples, and in non-smokers. We also examined the correlation structure between all derived and external MRSs using a heatmap summarizing their pairwise Pearson's correlation coefficient. Then, we compared the mean difference of each MRS score among smoking history using an ANOVA *F*-test and two-sample *t*-test to understand whether there was a dosage dependence in the cord blood DNAm signature of maternal smoking. Additionally, each score was tested against a binary outcome for current smoker vs. not, and two continuous measures for smoking history and weekly smoking exposure. The binary outcome was tested using a logistic regression model and the predictive performance was assessed using the area under the receiver operating characteristic curve (AUC). The

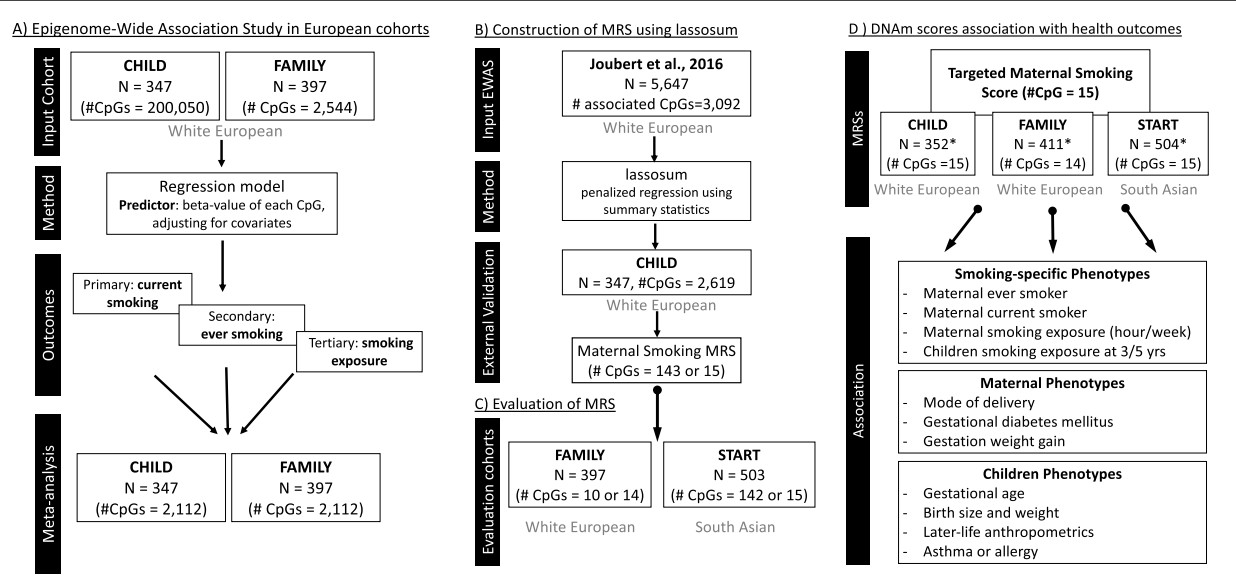

**Figure 1.** Schematic overview of the analytical pipeline for the cord blood DNA methylation (DNAm) maternal smoking score and association study. (**A**) shows the epigenome-wide association studies conducted in the European cohorts (CHILD and FAMILY); (**B**) illustrates the workflow for methylation risk score (MRS) construction using an external epigenome-wide association studies (EWAS) (*Joubert et al., 2016*) as the discovery sample and The Canadian Healthy Infant Longitudinal Development (CHILD) study as the external validation study, while (**C**) demonstrates the evaluation of the MRS in two independent cohorts of White European (i.e. FAMILY) and South Asian (i.e. START). The validated MRS was then tested for association with smoking-specific, maternal, and children phenotypes in CHILD, FAMILY, and START, as shown in (**D**). *indicates cohort sample size including those with missing smoking history.

reported 95% confidence interval for each estimated AUC was derived using 2000 bootstrap samples. The continuous outcome was examined using a linear regression model and its performance was quantified using the adjusted $R^2$.

For the derived MRS, we empirically assessed whether a systematic difference existed in the resulting score with respect to all other derived scores. This was examined via pairwise mean differences between the HM450 and other scores using a two-sample *t*-test and an overall test of mean difference using an ANOVA *F*-test, among all samples and the subset of never-smokers. Finally, we tested the association between each maternal smoking MRS and smoking phenotypes in mothers, as well as offspring phenotypes using a linear regression model, when applicable, adjusting for the child's age at each visit. The association results were meta-analyzed for phenotypes with homogeneous effects across the cohorts using a fixed-effect model. An FDR adjustment was used to control the multiple testing of meta-analyzed associations between MRS and 25 (or 23, depending on the number of phenotypes available in the cohort) outcomes, and we considered the association that passed an FDR-adjusted p-value <0.05 to be relevant.

## Results

### Cohort sample characteristics

The analyses included 763 European mother-child pairs with cord blood DNAm data from the CHILD study (CHILD; n=352)(*Subbarao et al., 2015*) and The Family Atherosclerosis Monitoring In earLY life (FAMILY; n=411) study (*Morrison et al., 2009*), and 503 South Asian mother-child pairs from The SouTh Asian biRth cohorT (START) study (*Anand et al., 2013*). A schematic overview of the analytical flow of the study can be found in *Figure 1*.

We observed lower past smoking and missingness on smoking history among pregnant women in START as compared to CHILD or FAMILY using the epigenetic subsample (*Table 1*) and the overall sample (*Supplementary file 1b*). Pregnant women in START were significantly different from CHILD or FAMILY in that they were on average younger at delivery, had a lower BMI, and a higher rate of GDM, in line with other cohort studies in South Asian populations (*Brydon et al., 2000*; *Farrar et al., 2015*).

As compared to START, newborn infants from CHILD and FAMILY had a longer gestational period, a higher birth weight, and a higher BMI at birth (*Table 1*; *Supplementary file 1b*). We observed no difference between cohorts in terms of parity or newborn sex in the epigenetic subsample (*Table 1*). However, self-reported smoking exposure, measured by the number of hours exposed to cigarette smoking per week, was highly skewed and zero-inflated across the three cohorts (*Figure 2—source data 1*).

Within the European epigenetic subsample, of the 744 mother–newborn pairs with complete smoking history data, 40 (5.3%) newborns were exposed to current maternal smoking, which is on the lower end of the spectrum for the prevalence of smoking during pregnancy (9.2–32.5%) among Canadians (*Lange et al., 2018*). In addition, mothers who smoked during pregnancy were on average younger, had fewer years of education, and had higher household exposure to smoking (*Supplementary file 1e*). However, there was no statistically significant difference between newborns exposed to current and none or previous smoking in terms of birth weight, birth length, gestational age, or estimated cord blood cell proportions.

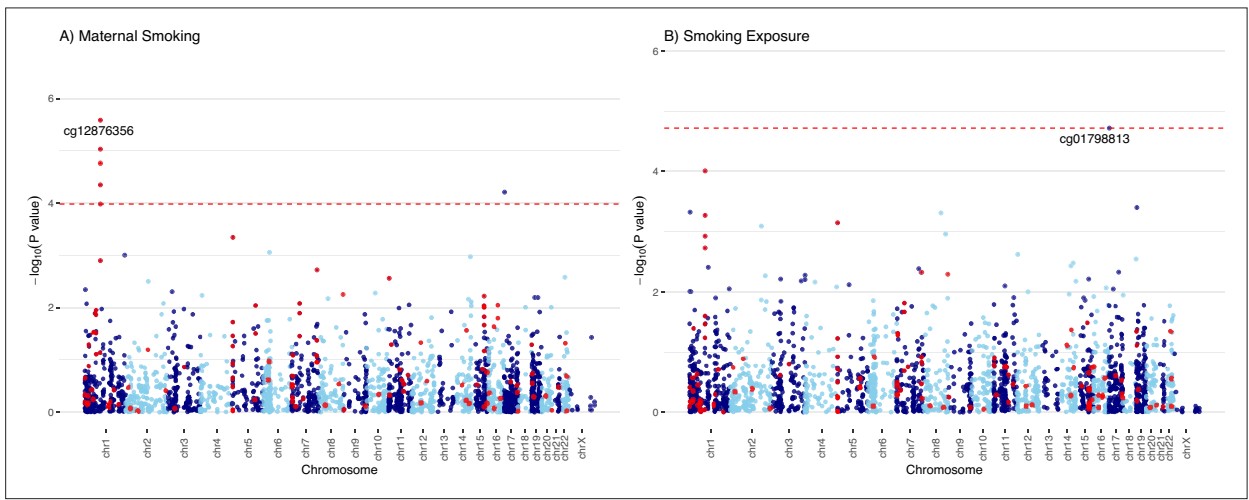

**Figure 2.** Manhattan plots of the meta-analyzed association between cord blood DNA methylation (DNAm) and maternal smoking in Europeans. Manhattan plots summarized the meta-analyzed association *p*-values between cord blood DNA methylation levels and current maternal smoking (**A**; n = 744) or smoking exposure (**B**; n = 735) at a common set of 2114 cytosine–phosphate–guanine (CpG) sites. The red line denotes the smallest -log10(*p*-value) that is below the false discovery rate (FDR) correction threshold of 0.05. The red dots represent established associations with maternal smoking reported by Joubert and colleagues (*Joubert et al., 2016*).

The online version of this article includes the following source data and figure supplement(s) for figure 2:

**Source data 1.** Histogram of the smoking exposure across the three cohorts.

**Figure supplement 1.** Manhattan plots of the meta-analyzed association between cord blood DNA methylation and ever maternal smoking in the combined European cohorts.

**Figure supplement 2.** Quantile-quantile plots of the meta-analyzed association between cord blood DNA methylation and maternal smoking history, smoking exposure in the combined European cohorts.

**Figure supplement 3.** Regression diagnostic for association between the top cytosine–phosphate–guanine (CpG) (cg09935388) and smoking exposure (n=339) without data transformation in The Canadian Healthy Infant Longitudinal Development (CHILD).

**Figure supplement 4.** Regression diagnostic for association between the top cytosine–phosphate–guanine (CpG) (cg09935388) and smoking exposure (n=396) without data transformation in Family Atherosclerosis Monitoring In early life (FAMILY).

**Figure supplement 5.** Regression diagnostic for association between the top cytosine–phosphate–guanine (CpG) (cg09935388) and smoking exposure (n=339) under an inverse normal rank transformation in Canadian Healthy Infant Longitudinal Development (CHILD).

**Figure supplement 6.** Regression diagnostic for association between the top cytosine–phosphate–guanine (CpG) (cg09935388) and smoking exposure (n=396) under an inverse rank transformation in Family Atherosclerosis Monitoring In early life (FAMILY).

**Figure supplement 7.** Scatterplots of meta-analyzed association effects for maternal smoking history or smoking exposure and reported effects of maternal smoking.

**Figure supplement 8.** Manhattan plots of the Epigenome-wide associations between cord blood DNA methylation (DNAm) and maternal smoking history, smoking exposure in Canadian Healthy Infant Longitudinal Development (CHILD).

## Epigenetic association of maternal smoking in White Europeans

The two predominantly White European cohorts, FAMILY (n=397) and CHILD (n=347), contributed to the meta-analysis of maternal smoking for both the primary outcome of current smoking (*Figure 1A*; *Figure 2A*) and the secondary outcome of ever smoking (*Figure 2—figure supplement 1*). The top associated CpGs with current maternal smoking were mapped to the growth factor independent 1 (*GFI1*) gene on chromosome 1, with cg12876356 as the lead (meta-analyzed effect = −1.11±0.22; meta-analyzed p=2.6 × 10$^{-6}$; FDR adjusted p=0.006; *Table 2*). There were no CpGs associated with the ever-smoker status at an FDR of 0.05, though the top signal (cg09935388) was also mapped to the *GFI1* gene (Pearson's r$^2$ correlation with cg12876356=0.75 and 0.68 in CHILD and FAMILY, respectively; *Figure 2—figure supplement 1*). The top associated CpG from the meta-analysis of smoking exposure (hours per week) in the European-origin cohorts (*Figure 2B*) was cpg01798813 on chromosome 17, which was also associated with maternal smoking and was consistent in the direction of association (meta-analyzed effect = −0.18±0.04; meta-analyzed p=1.4 × 10$^{-5}$; FDR adjusted p=0.04; *Table 2*). There was no noticeable inflation of empirical type I error in the association *p*-values from the meta-analysis, with the median of the observed association test statistic roughly equal to the expected median (*Figure 2—figure supplement 2*).

As a sensitivity analysis, we repeated the analysis for the continuous smoking exposure under rank transformation vs. raw phenotype for the associated CpG in *GFI1* and examined the regression diagnostics (*Figure 2—figure supplements 3–6*), and found that the model under rank-transformation deviated less from assumptions. Furthermore, we observed consistency in the direction of association for the 128 CpGs that overlapped between our meta-analysis and the 2620 CpGs with evidence of association for maternal smoking (*Joubert et al., 2016*; *Figure 2—figure supplement 7*). Specifically, the Pearson's correlation coefficient for maternal smoking and weekly smoking exposure was 0.72 and 0.60, respectively. The maternal smoking and smoking exposure EWASs in CHILD alone did not yield any CpGs after FDR correction (*Figure 2—figure supplement 8*).

## MRS captures maternal smoking and smoking exposure

The final MRSs, validated using CHILD European samples (n=347), included 15 and 143 CpG markers (*Supplementary file 1g*) from the targeted array and the epigenome-wide HM450 array (*Figure 1B*), respectively. Both produced methylation scores that were significantly associated with maternal smoking history (ANOVA F-test *p*-values = 1.0 × 10$^{-6}$ and 2.4×10$^{-14}$ in CHILD and 3.6×10$^{-16}$ and <2.2 × 10$^{-16}$ in FAMILY; *Figure 3*, *Figure 3—figure supplement 1*), and the best among alternative scores for CHILD and FAMILY (*Supplementary file 1f*). With the exception of the air pollution MRS, which only contained 3 CpGs (*Supplementary file 1f*), all remaining scores were marginally associated with smoking history in both CHILD and FAMILY (*Figure 3—figure supplement 1*) and correlated with each other (*Figure 3—figure supplement 2*). In particular, scores that were derived using the Joubert EWAS as the discovery sample, including ours, had higher pairwise correlation coefficients across the birth cohorts, with many of the CpGs mapping to the same genes, such as *AHRR*, *MYO1G*, *GFI1*, *CYP1A1*, and *RUNX3*. There was no statistically significant difference in mean between the two scores in any of the three cohorts (two-sample t-test *p*s >0.6) or among non-smokers (two-sample t-test *p*s >0.6; *Figure 3—figure supplement 3*). Since the HM450 score provides statistically more significant results in both CHILD and FAMILY with smoking history, despite the reduction in CpGs included (only 26 out of 143 CpGs present in FAMILY; *Supplementary file 1f*), we proceeded with the HM450 MRS model constructed using the 143 CpGs in subsequent analyses.

The HM450 MRS was significantly associated with maternal smoking history in CHILD (n=347) and FAMILY (n=397), but we failed to meaningfully validate the association in START (n=503) – not surprisingly – due to the low number of ever-smokers (n=2). A weak dose-dependent relationship between the MRS and the four categories of maternal smoking status in the severity of exposure ([0]=never smoked; [1]=quit before this pregnancy; [2]=quit during this pregnancy; [3]=currently smoking) was present in CHILD but was not replicated in FAMILY (*Figure 3*). The AUC for detecting current smokers were 0.95 (95% confidence interval: 0.89–1) and 0.89 (95% CI: 0.83–0.94) in CHILD and FAMILY (*Figure 3*), respectively, while the AUCs for detecting ever-smokers were 0.61 (95% CI: 0.54–0.67), 0.60 (95% CI: [0.55,0.69]; *Supplementary file 1f*), and 0.82 (95% CI: [0.55,1]; *Figure 3*), respectively. As a result, the epigenetic maternal smoking score was strongly associated with smoking status during pregnancy (OR = 1.09, 95% CI: [1.07,1.10], p=1.96 × 10$^{-32}$) in the combined European

**Table 2.** Meta-analysis results of the association between cytosine–phosphate–guanines (CpGs) and maternal smoking and smoking exposure that passed a marginal p<0.05 threshold after the false discovery rate correction in European cohorts.

| | CHR | Position | CpG | UCSC reference gene | Meta-analysis (CHILD and FAMILY) | | | | | Cohort-specific association P-value | | Reported Association EWAS catalog |
| | | | | | Fixed effect | Standard error | Association p-value | p-value for effect heterogeneity | FDR adjusted the Association P-value | CHILD | FAMILY | |
|---|---|---|---|---|---|---|---|---|---|---|---|---|
| | 1 | 92481269 | cg12876356 | GFI1 | –1.11 | 0.22 | 7.33E-07 | 0.51 | 0.0019 | 0.02 | 9.45E-06 | MS;S; AC; BW |
| | 1 | 92482032 | cg09935388 | GFI1 | –1.15 | 0.24 | 2.26E-06 | 0.52 | 0.0029 | 0.02 | 2.71E-05 | MS;GA; S; AC; BMI; BW |
| | 1 | 92482405 | cg14179389 | GFI1 | –1.48 | 0.32 | 5.03E-06 | 0.73 | 0.0035 | 0.01 | 1.12E-04 | MS;S |
| | 1 | 92481144 | cg18146737 | GFI1 | –0.92 | 0.20 | 5.58E-06 | 0.50 | 0.0035 | 0.04 | 3.95E-05 | MS;S; AC; BW |
| | 1 | 92480576 | cg09662411 | GFI1 | –0.94 | 0.22 | 1.64E-05 | 0.29 | 0.0083 | 0.10 | 3.85E-05 | MS;S |
| Maternal Smoking | 1 | 92481479 | cg18316974 | GFI1 | –0.74 | 0.18 | 3.58E-05 | 0.33 | 0.0152 | 0.13 | 7.34E-05 | MS;S; AC; BW |
| | 17 | 2494783943 | cg01798813 | – | –0.83 | 0.21 | 1.09E-04 | 0.34 | 0.0395 | 0.02 | 0.0016 | A; GA; BMI |
| Smoking Exposure | 1 | 92482032 | cg09935388 | GFI1 | –0.18 | 0.04 | 1.39E-05 | 0.23 | 0.04 | 0.15 | 2.45E-05 | MS;GA; S; AC; BMI; BW |
| | 17 | 2494783943 | cg01798813 | – | –0.18 | 0.04 | 3.30E-05 | 0.13 | 0.04 | 0.00035 | 0.013 | A; GA; BMI |

MS: maternal smoking. GA: gestational age. AC: alcohol consumption. BMI: body mass index. T2D: type 2 diabetes. A: age. BW: birth weight.

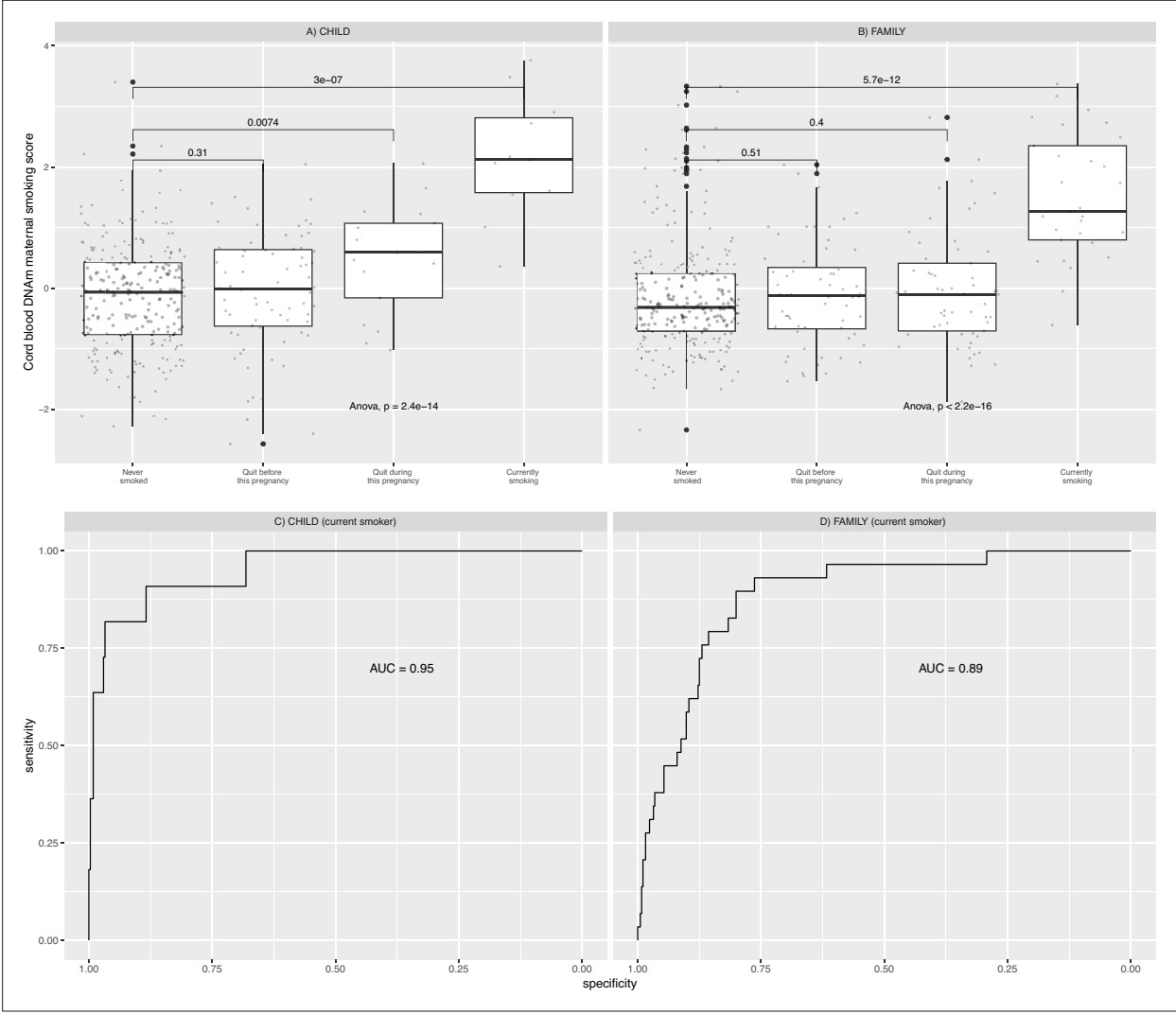

**Figure 3.** Relationships between maternal smoking methylation risk score (MRS) and maternal smoking history categories for Canadian Healthy Infant Longitudinal Development (CHILD) and Family Atherosclerosis Monitoring In early life (FAMILY). Maternal smoking methylation score (y-axis) was shown as a function of maternal smoking history (x-axis) in levels of severity for prenatal exposure for CHILD (**A**; n=347), and FAMILY (**B**; n=397). Each severity level was compared to the never-smoking group and the corresponding two-sample *t*-test *p*-value was reported. The analysis of variance via an F-test *p*-value was used to indicate whether a mean difference in methylation score was present among all smoking history categories. The area under the receiver operating characteristic curve (AUC) for each study was shown in the lower panel.

The online version of this article includes the following figure supplement(s) for figure 3:

**Figure supplement 1.** A comparison of results for derived and external maternal smoking methylation risk scores (MRSs).

**Figure supplement 2.** A heatmap of correlation between derived and external maternal smoking methylation risk score (MRSs).

**Figure supplement 3.** Comparison of all methylation scores stratified by study.

cohorts. Meanwhile, the maternal smoking MRS was significantly associated with increased number of hours exposed to smoking per week in the two White European cohorts (1.93±0.33 hr per 1 unit of increase in MRS, FDR adjusted p=1.2 × $10^{-7}$; *Supplementary file 1h*; cohort-specific p=5.4 × $10^{-5}$ in CHILD and p=2.3 × $10^{-5}$ in FAMILY; *Table 3*), but not in the South Asian birth cohort (p=0.58; *Table 3*).

Among individuals who had never smoked, no statistically significant mean difference was observed in the distribution of the combined methylation score between South Asian and European cohorts (*Supplementary file 1i*). These results provided empirical support for the portability of an European-derived maternal smoking methylation score to South Asian populations.

**Table 3.** Significant associations between maternal smoking methylation risk score and phenotypes in CHILD, FAMILY, and START.

| | CHILD | | | FAMILY | | | START | | |
|---|---|---|---|---|---|---|---|---|---|
| | Fixed effect | Standard error | Association p-value | Fixed effect | Standard error | Association p-value | Fixed effect | Standard error | Association P-value |
| Smoking exposure (hr/week) | 1.64 | 0.40 | 5.40E-05 | 2.58 | 0.60 | 2.34E-05 | 0.07 | 0.12 | 0.58 |
| 1 year Smoking exposure (hr/week) | 0.44 | 0.15 | 0.0044 | – | – | – | – | – | – |
| 3 year Smoking exposure (hr/week) | – | – | – | 1.15 | 0.39 | 0.0033 | – | – | – |
| Gestational weight gain (kg) | –0.36 | 0.38 | 0.35 | –0.62 | 0.26 | 0.017 | –0.14 | 0.34 | 0.69 |
| Gestational age (weeks) | 1.64 | 0.40 | 6.32E-05 | 2.84 | 0.62 | 5.52E-06 | 0.07 | 0.12 | 0.59 |
| Birth weight (kg) | –0.06 | 0.03 | 0.016 | –0.04 | 0.02 | 0.096 | –0.03 | 0.02 | 0.094 |
| Birth length (cm) | –0.14 | 0.15 | 0.35 | –0.10 | 0.10 | 0.33 | –0.37 | 0.12 | 0.0023 |
| 1 year Height (cm) | –0.32 | 0.16 | 0.047 | –0.34 | 0.14 | 0.019 | –0.42 | 0.16 | 0.0079 |
| 2 year Height (cm) | –0.13 | 0.35 | 0.72 | –0.26 | 0.17 | 0.14 | –0.57 | 0.21 | 0.0067 |
| 5 year Height (cm) | –0.36 | 0.26 | 0.16 | –0.43 | 0.26 | 0.095 | –0.47 | 0.37 | 0.21 |
| 3 year Skinfold thickness | 0.48 | 0.19 | 0.014 | 0.94 | 0.26 | 3.46E-04 | 0.24 | 0.27 | 0.38 |
| 5 year Skinfold thickness | 0.56 | 0.24 | 0.019 | 0.68 | 0.37 | 0.068 | 0.12 | 0.42 | 0.77 |

## Association between MRS and other phenotypes

We observed several notable associations with children outcomes (*Figure 1C*). The maternal smoking MRS was consistently associated with increasing weekly smoking exposure in children reported by mothers at the 1 year visit (0.44±0.15, p=0.0044; *Table 3*) in CHILD, and at 3 year visit (0.86±0.26, p=0.0037; *Table 3*) in FAMILY, but not in START as all mothers reported non-exposure to smoking in children. A higher maternal smoking MRS was significantly associated with smaller birth size (–0.37±0.12, p=0.0023; *Table 3*) and height at 1, 2, and 5- year visits in the South Asian cohort (*Table 3*). We observed similar associations with body size in the White European cohorts (heterogeneity p-values >0.2), collectively, the MRS was associated with a smaller birth size (–0.22±0.07, p=0.0016; FDR adjusted p=0.019; *Supplementary file 1h*) in the combined European and South Asian cohorts. Meanwhile, a higher maternal smoking MRS was also associated with a lower birth weight (–0.043±0.013, p=0.001; FDR adjusted p=0.011; *Supplementary file 1h*) in the combined sample, though the effect was weaker in START (–0.03±0.02; p=0.094; *Table 3*) as compared to the White European cohorts.

The meta-analysis revealed no heterogeneity in the direction nor the effect size of associations for body size and weight between populations at birth or at later visits (heterogeneity p-values = 0.16–1; *Supplementary file 1h*). The association between the MRS and several children phenotypes, including height or length, weight, and skinfolds, appeared to persist with similar estimated effects throughout early developmental years (*Supplementary file 1h*), albeit the most significant effects were at birth, and the significance attenuated at later visits. We did not find any association with self-reported allergy or asthma in children at later visits (*Supplementary file 1h*). Furthermore, there was no evidence of the association between the MRS and any maternal outcomes (*Supplementary file 1h*).

## Discussion

We examined the epigenetic signature of maternal smoking and smoking exposure using newborn cord blood samples from predominately European-origin and South Asian cohorts via two strategies:

an individual CpG-level EWAS approach, and a multivariate approach in the form of a methylation score. The EWAS results replicated the association between maternal smoking and CpGs in the *GFI1* gene that is well described in the literature with respect to smoking (*Joehanes et al., 2016*; *Sikdar et al., 2019*), maternal smoking (*Joubert et al., 2016*; *Hannon et al., 2019*; *Markunas et al., 2014*; *Richmond et al., 2015*), and birth weight (*Küpers et al., 2015*). In the latter case, we observed a significant association with maternal smoking history and smoking exposure in European-origin newborns. Furthermore, we noted a weak dose-dependent relationship between maternal smoking history and the methylation score in one European cohort (CHILD) but this was not replicated in the other (FAMILY). Since the timing and duration of maternal smoking during pregnancy were not directly available, these differences could play a role in the magnitude and specificity of DNA methylation changes in cord blood. Finally, the significant association of the MRS with the newborn health metrics in START, in the absence of mothers' active smoking, could be the result of underreporting of smoking, poor recall of the time of quitting, and/or due to air pollution exposure (*Rider and Carlsten, 2019*), leading to oxidative stress. This suggests that our cord blood DNAm signature of maternal smoking is perhaps not unique to cigarette smoking, but captures similar biochemical responses, for example, via the aryl hydrocarbon receptor (*Vogel et al., 2020*; *Reynolds et al., 2015*). Our observation that a higher MRS was associated with lower birth weight and smaller birth length in both ethnic populations is thus consistent with the established link between oxidative stress and metabolic syndrome (*Roberts and Sindhu, 2009*).

Contrary to DNA methylation studies of smoking in adults, where whole blood is often used as a proxy tissue, there are multiple relevant tissues for maternal smoking during pregnancy, including the placenta of the mother, newborn cord blood, and children's whole blood. However, methylation changes measured in whole blood or placenta of the mother, or cord blood of infants showed substantially different patterns of association signals (*Everson et al., 2021*). There are several advantages of using a cord blood-based biomarker from the DoHaD perspective. Firstly, cord blood provides a direct reflection of the in-utero environment and fetal exposure to maternal smoking. Additionally, since cord blood is collected at birth, it eliminates potential confounding factors such as postnatal exposures that may affect maternal blood samples. Furthermore, studying cord blood DNAm allows for the assessment of epigenetic changes specifically relevant to the newborn, offering valuable information on the potential long-term health implications. Meanwhile, methylation signals are known to be tissue-specific, thus it would be of interest for future research to combine differential methylation patterns from all relevant tissue to assess the immediate and long-term effects of maternal smoking. Another direction to further this line of research is to explore postnatal factors that mitigate prenatal exposures, for example, breastfeeding, which has been shown to have a protective effect against maternal tobacco smoking (*Moshammer and Hutter, 2019*). Indeed, more research is necessary to understand the critical periods of exposure and the dose-response relationship between maternal smoking and cord blood DNA methylation changes. Ongoing efforts to monitor the offspring and collect data in the next decade are in progress to establish the long-term association between maternal smoking and cardio-metabolic health (*Morrison et al., 2009*; *Anand et al., 2013*). As such, the constructed MRS can facilitate future research in child health and will be included as part of the generated data for others to access.

The strengths of this report include ethnic diversity, and fine phenotyping in a prospective and harmonized way with follow-up at multiple early childhood stages. This work is the first major multi-ancestry study that utilizes methylation scores to study maternal smoking and examines their portability from European-origin populations to South Asians. The use of MRS, as compared to individual CpGs, is a powerful tool to systematically investigate the influence of DNA methylation changes and whether it has lasting functional consequences on health outcomes. Our results converge with previous findings that epigenetic associations of maternal smoking are associated with newborn health, and add to the small body of evidence that these relationships extend to non-European populations and that different ancestral populations can experience the early developmental periods differently.

A few limitations should be mentioned. In the context of existing epigenetic studies of maternal smoking, we were not able to replicate signals in other well-reported genes such as *AHRR*, *CYP1A1*, and *MYO1G*, however, the MRS was able to pick up signals from these genes (*Supplementary file 1g*). This could be due to several reasons. First, the customized array with a limited number of CpGs (<3000) was designed in 2016 and many large EWASs on smoking and maternal smoking conducted

more recently had not been included. Nonetheless, we have shown that from a multivariate perspective, the MRS constructed using a targeted approach that was carefully designed can be equally powerful with the advantage of being cost-effective. Second, contrary to existing EWASs where the methylation values are typically treated as the outcome, and the exposure, such as smoking, as the predictor; we reversed the regression such that the methylation levels were the predictors and smoking exposure as the outcome. This reverse regression approach is robust and our choice to reverse the regression was motivated by the goal of constructing a smoking score that combines the additive effects at multiple CpGs, which would otherwise be unfeasible. Third, systematic ancestral differences in DNA methylation patterns had been shown to vary at individual CpGs in terms of their association with smoking (*Elliott et al., 2014*). Converging with this conclusion, we also found the association with *GFI1* to be most consistent after adjusting for cell composition. Fourth, while it would be of interest to examine a broader range of health outcomes in children, such as lung health and allergies, we were unable to acquire and standardize this information across different cohorts. This aspect should be considered in future study designs. Finally, maternal smoking is often associated with other confounding factors, such as socioeconomic status, other lifestyle behaviors, and environmental exposures. While we have done our best to control for well-known confounders that were available by study design, as in all observational studies, we could not account for unknown confounding effects. Finally, in recent years, maternal smoking has been on a decline as a result of changes in social norms and public health policies (*Martin et al., 2023*). This is also consistent with the lower smoking rates observed in our European cohorts (CHILD and FAMILY). Given the proportion of current smokers, the effective sample size for a direct comparison between CHILD and FAMILY, i.e., equivalently-powered sample size of a balanced (50% cases, 50% controls) design, were 41.7 and 104.7, respectively. While CHILD had a lower effective sample size, we ultimately chose it for validating the methylation score to better cover the CpGs that were significant in the discovery of EWAS. A larger validation study will likely further boost the performance of the methylation score and be considered in future research.

In conclusion, the epigenetic maternal smoking score we constructed was strongly associated with smoking status during pregnancy and self-reported smoking exposure in White Europeans, and with smaller birth size and lower birth weight in the combined South Asian and White European cohorts. The proposed cord blood epigenetic signature of maternal smoking has the potential to identify newborns who were exposed to maternal smoking in utero and to assess the long-term impact of smoking exposure on offspring health. In South Asian mothers with minimal smoking behavior, the relationship between the methylation score and negative health outcomes in newborns is still apparent, indicating that DNA methylation response is sensitive to smoking exposure, even in the absence of active smoking.

## Acknowledgements

We express our sincere gratitude to all the participating families and the START, FAMILY, and CHILD study teams, including interviewers, nurses, computer and laboratory technicians, clerical workers, research scientists, volunteers, managers, and receptionists.

We would like to acknowledge the Genetic and Molecular Epidemiology Laboratory (GMEL), an associate of Hamilton Health Sciences and McMaster University, for their indispensable contributions to this work. The technical staff of GMEL conducted all epigenetic profiling, including sample processing and other technical operations.

We thank the members of the Nutrigen Alliance for providing the data: Sonia S Anand; Stephanie A Atkinson; Meghan Azad; Allan B Becker; Jeffrey Brook; Judah A Denburg; Dipika Desai; Russell J de Souza; Milan K Gupta; Michael Kobor; Diana L Lefebvre; Wendy Lou; Piushkumar J Mandhane; Sarah McDonald; Andrew Mente; David Meyre; Theo J Moraes; Katherine M Morrison; Guillaume Paré; Malcolm R Sears; Padmaja Subbarao; Koon K Teo; Stuart E Turvey; Julie Wilson; Salim Yusuf; Gita Wahi; Michael A Zulyniak.

This study was funded by the Canadian Institutes of Health Research Metabolomics Team Grant: MWG-146332. Dr. Anand is supported by a Tier 1 Canada Research Chair in Ethnicity and CVD and Heart, Stroke Foundation Chair in Population Health, a grant from the Canadian Partnership Against Cancer, Heart and Stroke Foundation of Canada and Canadian Institutes of Health Research. Dr. Azad is supported by a Tier 2 Canada Research Chair in the Developmental Origins of Chronic Disease.

# Additional information

## Funding

| Funder | Grant reference number | Author |
|---|---|---|
| Canadian Institutes of Health Research | MWG-146332 | Russell J de Souza<br>Katherine M Morrison<br>Stephanie A Atkinson<br>Padmaja Subbarao<br>Koon K Teo<br>Guillaume Paré<br>Sonia S Anand |
| Canadian Institutes of Health Research | Tier 1 Canada Research Chair in Ethnicity and CVD and Heart, Stroke Foundation Chair in Population H | Sonia S Anand |

The funders had no role in study design, data collection and interpretation, or the decision to submit the work for publication.

## Author contributions

Wei Q Deng, Conceptualization, Formal analysis, Validation, Investigation, Visualization, Methodology, Writing – original draft, Writing – review and editing; Nathan Cawte, Data curation, Validation, Methodology, Writing – review and editing; Natalie Campbell, Data curation, Validation, Investigation, Project administration, Writing – review and editing; Sandi M Azab, Validation, Investigation, Project administration, Writing – review and editing; Russell J de Souza, Funding acquisition, Validation, Investigation, Writing – review and editing; Amel Lamri, Validation, Investigation, Writing – review and editing; Katherine M Morrison, Stephanie A Atkinson, Padmaja Subbarao, Koon K Teo, Data curation, Funding acquisition, Validation, Investigation, Writing – review and editing; Stuart E Turvey, Theo J Moraes, Piush J Mandhane, Meghan B Azad, Elinor Simons, Data curation, Validation, Investigation, Writing – review and editing; Guillaume Paré, Conceptualization, Resources, Data curation, Funding acquisition, Validation, Investigation, Methodology, Writing – review and editing; Sonia S Anand, Conceptualization, Resources, Data curation, Funding acquisition, Validation, Investigation, Project administration, Writing – review and editing

## Author ORCIDs

Wei Q Deng ⓘ https://orcid.org/0000-0003-4212-2607
Natalie Campbell ⓘ https://orcid.org/0009-0004-2983-1542
Guillaume Paré ⓘ https://orcid.org/0000-0002-6795-4760
Sonia S Anand ⓘ https://orcid.org/0000-0003-3692-7441

## Ethics

Human subjects: Ethical approval was obtained independently from the Hamilton Integrated Research Ethics Board: CHILD (REB 07-2929), FAMILY (REB 02-060), and START (REB 10-640). CHILD was additionally approved by the respective Human Research Ethics Boards at McMaster University, the Universities of Manitoba, Alberta, and British Columbia, and the Hospital for Sick Children. Legal guardians of each participant provided written informed consent. Written informed consent was obtained from the parent/guardian (participating mother) for each study separately. We also have now obtained additional ethics board approval from HiREB (REB 16592) for using the data from the three cohorts together without additional consent from the participants.

Reviewer #2 (Public Review): https://doi.org/10.7554/eLife.93260.4.sa1
Reviewer #3 (Public Review): https://doi.org/10.7554/eLife.93260.4.sa2
Author response https://doi.org/10.7554/eLife.93260.4.sa3

## Additional files

### Supplementary files

• Supplementary file 1. Additional tables and summaries of results. (A) Quality controls for the inclusion/exclusion of samples and methylation probes. (B) Characteristics of the overall sample include 5176 mother–newborn pairs from the Canadian Healthy Infant Longitudinal Development (CHILD), Family Atherosclerosis Monitoring In early life (FAMILY), and SouTh Asian biRth cohorT (START) cohorts. (C) A summary of available analyses and outcome variables in each cohort. (D) A summary of the DNA methylation (DNAm) maternal smoking score derivation design and results. (E) Characteristics of the epigenetic subsample from CHILD and FAMILY cohorts stratified by smoking status. (F) Score weights for external DNAm maternal smoking scores. (G) summary of cytosine–phosphate–guanines (CpGs) that contribute to the DNAm maternal smoking scores and their weights. (H) Association between maternal smoking methylation risk score and phenotypes in CHILD, FAMILY, and START. (I) Summary of mean difference in methylation risk scores between studies in overall samples and those never smoked.

### Data availability

The summary statistics used to construct methylation risk scores are available from EWAS catalog at http://www.ewascatalog.org/?trait=maternal%20smoking%20in%20pregnancy with additional filters of PubMID 27040690 and analysis on "Sustained maternal smoking in pregnancy effect on newborns adjusted for cell composition". Summary statistics generated in the current study, including a total of 7 primary association studies (three smoking phenotypes in the two European cohorts and smoking exposure in the South Asian cohort) and 3 sets of meta-analyzed results in Europeans are available from the Zenodo repository (10.5281/zenodo.13286433). All scripts to reproduce and validate the predictive model can be found at https://github.com/WeiAkaneDeng/EpigeneticResearch/tree/WeiAkaneDeng-patch-1/MaternalSmoking (copy archived at *Deng, 2024*).

The following dataset was generated:

| Author(s) | Year | Dataset title | Dataset URL | Database and Identifier |
|---|---|---|---|---|
| Deng W, Anand S | 2024 | Maternal smoking DNA methylation risk score associated with health outcomes in offspring of European and South Asian ancestry | https://doi.org/10.5281/zenodo.13286433 | Zenodo, 10.5281/zenodo.13286433 |

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
