## [Editor Report · eLife assessment]

This study offers a **useful** advance by introducing a cord blood DNA methylation score for maternal smoking effects, with the inclusion of cohorts from diverse backgrounds. However, the overall strength of evidence is deemed **incomplete**, due to concerns regarding low exposure levels and low statistical power, which hampers the generalisability of their findings. The study provides an interesting basis for future studies, but would benefit from the addition of more cohorts to validate the findings and a focus on more diverse health outcomes.

---

## [Referee Report · Reviewer #2 (Public Review)]

Summary:

The authors generated a DNA methylation score in cord blood for detecting exposure to cigarette smoke during pregnancy. They then asked if it could be used to predict height, weight, BMI, adiposity and WHR throughout early childhood.

Strengths:

The study included two cohorts of European ancestry and one of South Asian ancestry.

Weaknesses:

(1) Numbers of mothers who self-reported any smoking was very low likely resulting in underpowered analyses.

(2) Although it was likely that some mothers were exposed to second-hand smoke and/or pollution, data on this was not available.

(3) One of the European cohorts and half of the South Asian cohort had DNA methylation measured on only 2500 CpG sites including only 125 sites previously linked to prenatal smoking.

---

## [Referee Report · Reviewer #3 (Public Review)]

Summary:

Deng et al. assess neonatal cord blood methylation profiles and the association with (self-reported) maternal smoking in multiple populations, including two European (CHILD, FAMILY) and one South Asian (START), via two approaches: (1) they perform an independent epigenome-wide association study (EWAS) and meta-analysis across the CHILD and FAMILY cohort, during which they also benchmark previously reported maternal-smoking associated sites, and (2) they generate new composite methylation risk scores for maternal smoking, and assess their performance and association with phenotypic characteristics in the three populations, in addition to previously described maternal smoking methylation risk scores.

Strengths and weaknesses:

Their meta-analysis across multiple cohorts and comparison with previous findings represents a strength. In particular the inclusion of a South Asian birth cohort is commendable as it may help to bolster generalizability. However, their conclusions are limited by several important weaknesses:

(1) the low number of (self-reported) maternal smokers in particular their South Asian population, resulting in an inability to conduct benchmarking of maternal smoking sites in this cohort. As such, the inclusion of the START cohort in certain figures is not warranted (e.g., Figure 3) and the overall statement that smoking-associated MRS are portable across populations are not fully supported;

(2) different methylation profiling tools were used: START and CHILD methylation profiles were generated using the more comprehensive 450K array while the FAMILY cohort blood samples were profiled using a targeted array covering only 3,000, as opposed to 450,000 sites, resulting in different coverage of certain sites which affects downstream analyses and MRS, and importantly, omission of potentially relevant sites as the array was designed in 2016 and substantial additional work into epigenetic traits has been conducted since then;

(3) the authors train methylation risk scores (MRS) in CHILD or FAMILY populations based on sites that are associated with maternal smoking in both cohorts and internally validate them in the other cohort, respectively. As START cohort due to insufficient numbers of self-reported maternal smokers, the authors cannot fully independently validated their MRS, thus limiting the strength of their results.

Overall strength of evidence and conclusions:

Despite these limitations, the study overall does explore the feasibility of using neonatal cord blood for the assessment of maternal smoking. However, their conclusion on generalizability of the maternal smoking risk score is currently not supported by their data as they were not able to validate their score in a sufficiently large number of maternal smokers and never smokers of South Asian populations.

While their generalizability remains limited due to small sample numbers and previous studies with methylation risk scores exist, their findings may nonetheless provide the basis for future work into prenatal exposures which will be of interest to the research community. In particular their finding that the maternal smoking-associated MRS was associated with small birth sizes and weights across birth cohorts, including the South Asian birth cohort that had very few self-reported smokers, is interesting and the author suggest these findings could be associated with factors other than smoking alone (e.g., pollution), which warrant further investigation and would be highly novel.

Future exploration should also include a strong focus on more diverse health outcomes, including respiratory conditions that may have long-lasting health consequences.

---

## [Author Response]

The following is the authors’ response to the previous reviews.

**Reviewer #2:**
(1) P-values should be reported adjusted for multiple tests or, at the very least, note that they are unadjusted to alert the reader that they may be biased by winner's curse.

Throughout the manuscript, we applied the false discovery rate threshold to declare results that were statistically relevant for discussion. However, for reporting in abstract, we believe the raw p-values are most straightforward as we only reported the most important and robust results, and considering that (1) multiple testing correction does not change the ranking of the adjusted p-values; (2) p-value adjustment depends on both the method and the number of hypothesis tested; (3) all reporting of the most significant discovery results are prone to winner’s curse, but in the context of our study: the GFI1 finding was confirmatory in nature, thus raw p-value allows for a direct comparison with existing studies.

We have taken the suggestion to quote the FDR-adjusted p-values throughout the manuscript for meta-analyzed results and discussed the impact of FDR correction for the EWAS and MRS association differed as a result of the number of hypothesis in each context:

“For each EWAS or meta-analysis, the false discovery rate (FDR) adjustment was used to control multiple testing and we considered CpGs that passed an FDR-adjusted p-value < 0.05 to be relevant for maternal smoking.”

“An FDR adjustment was used to control the multiple testing of meta-analyzed association between MRS and 25 (or 23, depending on the number of phenotypes available in the cohort) outcomes, and we considered association that passed an FDR-adjusted p-value < 0.05 to be relevant.”

(2) The odds ratios and p-values reported in the abstract for associations of the MRS with smoking status and smoking exposure per week appear to be missing from the results section of the manuscript or (supplementary) tables.

The results for smoking status during pregnancy was added to the results:

“As a result, the epigenetic maternal smoking score was strongly associated with smoking status during pregnancy (OR=1.09 [1.07,1.10], p=5.5×10-33) in the combined European cohorts.”

The exposure association was reported in the result section and Supplementary Table 8. We do note the typo in the cohort specific p-values, which now has been corrected.

(3) It is misleading to report a lack of MRS associations with maternal smoking in South Asians without also stating that there were only two smokers.

We agree with the reviewer that an association test would not be justified given the lack of smoking in the present South Asian cohort. We also removed the p-value of association for the START cohort in Figure 3, based on this and comment #4 from reviewer #3. The relevant results have been revised as follows:

“The HM450 MRS was significantly associated with maternal smoking history in CHILD and FAMILY (n = 397), but we failed to meaningfully validate the association in START (n = 503; Figure 3) – not surprisingly – due to the low number of ever-smokers (n = 2).”

(4) It is potentially confusing to report MRS associations with maternal smoking by ethnicity but then report associations with birth size and length combined without any explanation. The most novel result of this study is that there is virtually no maternal smoking among the South Asians and yet the MRS is associated with birth weight and size and with height at age 2. This result is buried in the combined analysis. I would suggest reporting the MRS associations with height and weight separately as has been done for maternal smoking behavior.

We thank the reviewer for this suggestion and this has now been added the new Table 3, showing the cohort specific and meta-analyzed effect sizes. In the revision, we highlighted that the ethnic specific MRS associations, such as with smoking exposure at various age (1 and 3 years) and skinfold thickness in European cohorts but not the South Asian cohort, as well as associations that were more homogenous, such as the birth weight and unique body size association in combined cohorts. In particular, the MRS in the South Asian cohort exhibited a consistent association with body size at various time points (at birth, 1, 2, and 5 year) with similar effect sizes. The following was added to the results:

“A higher maternal smoking MRS was significantly associated with smaller birth size (-0.37±0.12, p = 0.0023; Table 3) and height at 1, 2, and 5 year visits in the South Asian cohort (Table 3). We observed similar associations with body size in the white European cohorts (heterogeneity p-values> 0.2), collectively, the MRS was associated with a smaller birth size (-0.22±0.07, p=0.0016; FDR adjusted p = 0.019) in the combined European and South Asian cohorts (Table 3). Meanwhile, a higher maternal smoking MRS was also associated with a lower birth weight (-0.043±0.013, p = 0.001; FDR adjusted p = 0.011) in the combined sample, though the effect was weaker in START (-0.03±0.02; p = 0.094) as compared to the white European cohorts.

The meta-analysis revealed no heterogeneity in the direction nor the effect size of associations for body size and weight between populations at birth or at later visits (heterogeneity p-values = 0.16–1; Supplementary Table 8).”

**Reviewer #3:**
(1) You mention that the 450K Score performs best even though only 10/143 are included for some populations. Did you explore recalibration of the MRS using only those 10 CpGs?

We thank the reviewer for this comment – due to an error in result transferring, the number of overlapping CpGs between the 450K score and the targeted array was in fact 26. This error only impacted results relevant to the FAMILY study using the HM450K score and did not materially change our results nor conclusions. We have updated accordingly, Table 3, Suppl. Tables 5, 8, 9, Figure 3-B, and Suppl. Figures 5, 6-B, 7-B and 7-D, and throughout the manuscript for meta-analyzed MRS associations.

The subset of 26 CpGs using the originally derived weight was expected to perform worse than the original HM450K score using the full 143 CpGs. When we did restrict the methylation score construction to these 26 CpGs, the performance in CHILD was worse than the original score, but comparable to FAMILY (updated Suppl. Table 5). These 26 CpGs did overlap with the targeted score derived in CHILD (13 out of 15 present) and in FAMILY (19 out of 63 present), suggesting moderate agreement between the array platform as well as across studies.

In other words, while the subset of 26 CpGs had reasonable performance in both CHILD and FAMILY, both studies could benefit by inclusion of the additional CpGs in the original score. We have included a sentence to discuss the choice of validation study and the trade-off between sample size and # of CpGs under response to Reviewer 3 comment # 2.

(2) Could the internal validation performance be driven by sample size of the training, providing support for the need for larger training sizes? Should this be discussed in the study?

The validation study, CHILD, has the smaller sample size between the two European cohorts. While both potential data for validation had smaller sample sizes, we chose CHILD (n=347), rather than FAMILY (n=397) as it had better coverage with respect to the discovery EWAS or the training data (# of associated CpGs = 3,092, n = 5,647). Beyond the signals of association, the validation performance also depends on a mix of overall sample size and the proportion of current smokers. Given the proportion of current smokers, the effective sample size for a direct comparison, i.e. equivalently-powdered sample size of a balanced (50% cases, 50% controls) design, are 41.7 and 104.7 for CHILD and FAMILY, respectively. While we are unable to directly compare whether a larger effective sample size produced a better performing score, we believe this to be the case, and thus a larger validation study would boost the performance of the methylation score. We have added the following to the discussion:

“Given the proportion of current smokers, the effective sample size for a direct comparison between CHILD and FAMILY, i.e. equivalently-powdered sample size of a balanced (50% cases, 50% controls) design, were 41.7 and 104.7, respectively. While CHILD had a lower effective sample size, we ultimately chose it for validating the methylation score to better cover the CpGs that were significant in the discovery EWAS. A larger validation study will likely further boost the performance of the methylation score and be considered in future research.”

(3) Figure 1: It is very helpful to have an overview diagram, but this should then follow the flow of the manuscript to aid the reader. Currently, the diagram does not follow the flow of the manuscript and thus is rather confusing - for instance, the figure starts with the MRS but initially an EWAS is conducted in the manuscript itself. I suggest to adapt the overview figure accordingly. Moreover, a description for (A), (B), (C) is not provided in the figure legends. Figure 1 could thus be improved further.

We thank the reviewer for the suggestion to improve the key figure that summarizes the manuscript. The EWAS workflow for the primary, secondary and tertiary outcomes, as well as the European cohorts meta-analysis has been added to the updated sub-figure A. The description for each subfigures has also been added to the figure legends as follows:

“Figure 1-A shows the epigenome-wide association studies conducted in the European cohorts (CHILD and FAMILY); Figure 1-B illustrated the workflow for methylation risk score (MRS) construction using an external EWAS (Joubert et al., 2016) as the discovery sample and CHILD study as the external validation study, while Figure 1-C demonstrates the evaluation of the MRS in two independent cohorts of white European (i.e. FAMILY) and South Asian (i.e. START). The validated MRS was then tested for association with smoking specific, maternal, and children phenotypes in CHILD, FAMILY, and START, as shown in Figure 1-D.”

(4) Figure 3: The readability and information content in this figure, and other figures containing boxplots (e.g., Supplementary Figure 5), could be improved. I would suggest to justify X axis labels to the axis rather than overlapping, and importantly, show individual data points wherever possible (e.g., overlaying the box plots). In (c), the ANOVA is not justified given the sample size in START. In general, it is worth excluding the START cohorts from this analysis on the justification of a too small sample size for maternal smokers.

We thank the reviewer for their thoughtful points for improvement. The axis labels have been wrapped to avoid overlapping, and the data points added to the boxplots. ANOVA p-value for START was removed due to the low counts of smokers in the figure and manuscript throughout. However, we retained START in Figure 3 and other boxplots to show the distribution of the score for non-smokers to benchmark with the European cohorts.

(5) In addition to boxplots, it may be helpful to show AUC diagrams for ROC curves (e.g. Figure 3). AUCs are reported in the Tables but not shown. Additionally, all AUC results should include 95% Confidence intervals.

This is a great suggestion and we have added the corresponding ROC, annotated with AUC (95% CI) to Figure 3. The 95% CI for all AUC results were added to the Tables and main text. The following was added to Methods:

“The reported 95% confidence interval for each estimated AUC was derived using 2,000 bootstrap samples.”

(6) Supplementary Figure 6: It could be helpful to discuss the amount of overlap between the different MRS.

Most of the scores were derived using the Joubert et al., (2016) EWAS as the discovery sample, including ours, and thus there will be overlap between the scores. The exception was the GondaliaScore, which contained only 3 CpGs that do not overlap with any other scores.

While different scores might not have selected completely identical sets of CpGs, the mapped genes are highly consistent across the scores. We have added to the discussion and results the extent of overlap between the top scores:

“In particular, scores that were derived using the Joubert EWAS as the discovery sample, including ours, had higher pairwise correlation coefficients across the birth cohorts, with many of the CpGs mapping to the same genes, such as AHRR, MYO1G, GFI1, CYP1A1, and RUNX3.”

(7) Supplementary Figure 7: This figure is never referenced in the text and from the legend itself it is not too clear what it is trying to show. Please refer to it in the main text with some additional context.

Supplementary Figure 7 was referenced in the Results under subsection “Methylation Risk Score (MRS) Captures Maternal Smoking and Smoking Exposure”, following the

Methods subsection “Statistical analysis” where we wanted to examine a systematic difference. We made revision to the main text to clarify the analysis:

“For the derived MRS, we empirically assessed whether a systematic difference existed in the resulting score with respect to all other derived scores. This was examined via pairwise mean differences between the HM450 and other score using a two-sample t-test and an overall test of mean difference using an ANOVA F-test, among all samples and the subset of never smokers.”

(8) Tables: Tables are currently challenging to read and perhaps more formatting could be done to improve readability.

We thank the reviewer for the suggestion. Main tables have been reformatted to a landscape layout and each numeric cell moved to the centre to improve readability.